# Protective Efficacy of Anti-Hyr1p Monoclonal Antibody against Systemic Candidiasis Due to Multi-Drug-Resistant *Candida auris*

**DOI:** 10.3390/jof9010103

**Published:** 2023-01-12

**Authors:** Shakti Singh, Ashley Barbarino, Eman G. Youssef, Declan Coleman, Teclegiorgis Gebremariam, Ashraf S. Ibrahim

**Affiliations:** 1Division of Infectious Diseases, The Lundquist Institute for Biomedical Innovation at Harbor-University of California at Los Angeles (UCLA) Medical Center, Torrance, CA 90502, USA; 2David Geffen School of Medicine, University of California, Los Angeles, CA 90095, USA; 3Biology Department, Pomona College, Pomona, CA 91711, USA

**Keywords:** *Candida auris*, candidiasis, immunotherapy, monoclonal antibody, Hyr1p

## Abstract

*Candida auris* is a multi-drug-resistant fungal pathogen that can survive outside the host and can easily spread and colonize the healthcare environment, medical devices, and human skin. *C. auris* causes serious life-threatening infections (up to 60% mortality) in immunosuppressed patients staying in such contaminated healthcare facilities. Some isolates of *C. auris* are resistant to virtually all clinically available antifungal drugs. Therefore, alternative therapeutic approaches are urgently needed. Using in silico protein modeling and analysis, we identified a highly immunogenic and surface-exposed epitope that is conserved between *C. albicans* hyphal-regulated protein (Cal-Hyr1p) and Hyr1p/Iff-like proteins in *C. auris* (Cau-HILp). We generated monoclonal antibodies (MAb) against this Cal-Hyr1p epitope, which recognized several clinical isolates of *C. auris* representing all four clades. An anti-Hyr1p MAb prevented biofilm formation and enhanced opsonophagocytic killing of *C. auris* by macrophages. When tested for in vivo efficacy, anti-Hyr1p MAb protected 55% of mice against lethal systemic *C. auris* infection and showed significantly less fungal burden. Our study is highly clinically relevant and provides an effective alternative therapeutic option to treat infections due to MDR *C. auris*.

## 1. Introduction

Multi-drug resistance is often associated with bacterial infections and rarely with fungal infections. However, this notion is now changing due to the rise in drug resistance in fungal pathogens such as *Candida auris* [1]. Discovered only a decade ago, *C. auris* has been reported from >140 countries around the globe [2,3]. *C. auris* evolved simultaneously mainly in four major geographical regions: South Asia (Clade I), East Asia (Clade II), Africa (Clade III), and South America (Clade IV) [1]. It has been established that 90% and 30% of the clinical isolates are resistant to at least one or two antifungal drugs, respectively, while several clinical isolates are resistant to all clinically available antifungal drugs and are therefore untreatable [4,5,6]. *C. auris* can easily spread and colonize inanimate objects and human skin in healthcare settings and poses a risk to immunosuppressed patients staying in such contaminated healthcare facilities [7,8,9]. *C. auris* can infect this patient population through attached invasive medical devices and surgical procedures [7,10,11,12,13,14,15]. Once *C. auris* gains access to the bloodstream, it can disseminate to the kidney, heart, and other target organs [16]. *C. auris* bloodstream infection is very difficult to treat and has a very high mortality rate of up to ~60% [1]. More recently, the widespread use of corticosteroid therapy to manage COVID-19-related infections resulted in the rise of COVID-19-associated fungal infections including *C. auris* [17,18,19,20]. Consequently, the U.S. Centers for Disease Control and Prevention (CDC) declared *C. auris* as an “urgent threat to public health” in its recent antimicrobial drug resistance report (AMR) [21,22,23]. 

*C. auris* has shown a wide range of drug resistance mechanisms (mutations in ERG genes involved in ergosterol synthesis, efflux pump upregulations, etc.) across all clades, thus making it difficult to develop new drug variants within the existing antifungal class (azoles, polyenes, and echinocandins) [1,24,25]. Therefore, newer antifungal interventions are urgently required. However, due to the multi-drug-resistant phenotype, *C. auris* will likely develop resistance to any new drug developed in the future. Thus, alternative immune therapeutic approaches using monoclonal antibodies can be more desirable in treating MDR pathogens including *C. auris*. A pathogen is less likely to develop resistance against monoclonal antibodies that target conserved epitopes and can be used in combination with existing antimicrobial drugs. 

*C. albicans* express several Glycosyl Phosphatidyl Inositol (GPI)-anchored cell wall proteins which are critical in the fungus attachment to host cells, biofilm formation, and iron acquisition [26,27]. Agglutinin-like sequence protein (Als) family and hyphal-regulated (Hyr/Iff)-like proteins are two of the major proteins which are known for adhesion/invasion of host tissues and evasion of host immune defense, respectively [28,29,30,31]. The N-terminal of these proteins is predominantly conserved and forms a β-helical structure followed by an α-crystallin domain and shows structural similarities with different bacterial adhesins [32,33,34]. 

Our research on *C. albicans* has shown tremendous success in developing alternative immune-based approaches to control fungal infection. We have identified two *C. albicans* cell surface proteins as potential vaccine candidates: Als3 protein (an adhesin/invasin) [35,36,37,38,39] and Hyr1p (a phagocyte evasion factor) [31]. Vaccines containing recombinant N-terminal region of Cal-Als3p or Cal-Hyr1p protected mice against systemic *C. albicans* infection [31,40]. Moreover, NDV-3A, a Cal-Als3p-based Alum adjuvanted vaccine, protected women of <40 years of age against vulvovaginal candidiasis in a phase 1b/2a clinical trial [41]. We also discovered Cal-Als3p orthologs in *C. auris* and showed that immunosuppressed mice can be protected by the NDV-3A vaccine from *C. auris* [42]. Furthermore, both Cal-Als3p and Cal-Hyr1p are cross-kingdom vaccine antigens that have cross-protective epitopes on *Staphylococcus aureus* and Gram-negative bacteria (*Acinetobacter baumannii*, *Klebsiella pneumoniae,* and *Pseudomonas aeruginosa*) [43,44,45], respectively. We determined that Cal-Hyr1p epitope#5 is responsible for this cross-reactive immunity against Gram-negative bacteria, and a monoclonal IgM antibody (MAb) targeting epitope#5 protected mice from pneumonia due to Gram-negative bacteria [44].

Recently, *C. auris* was shown to contain Cal-Hyr1/Iff-like proteins (Cau-HIL) [46]. Here, we identified a highly immunogenic and surface-exposed epitope (epitope#5) that is conserved between Cal-Hyr1p and Cau-HILp. An anti-Hyr1p epitope#5 monoclonal antibody can cross-recognize *C. auris* in vitro and protect mice from *C. auris* hematogenously disseminated infection. These findings are highly clinically relevant and provide an effective alternative to control infection by this MDR pathogen.

## 2. Materials and Methods

### 2.1. Candida Culture and Strains

*C. auris* strains representing all four clades (I–IV) were obtained from the Centers for Disease Control and Prevention (CDC, Atlanta, Courtesy of Dr. Shawn Lockhart). Both *C. albicans* (SC5314) and *C. auris* strains were grown in Yeast Extract Peptone Dextrose (YPD) broth overnight in a shaker incubator at 30 °C/200 rpm. The next day, the yeast cells were pelleted by centrifuging at 4000 rpm for 10 min at 4 °C, followed by washing three times with 1× phosphate buffered saline (PBS). The yeast cells were finally suspended in 1× PBS and counted by using a hemocytometer. For monoclonal antibody binding assays, 5 × 10^6^ cells/mL each of *C. albicans* and *C. auris* strains were added to RPMI-1640 media (supplemented with L-Glutamine and 10% fetal bovine serum) to allow germ tube formation (for *C. albicans* and mock conditions for *C. auris*) at 37 °C for 75 min with shaking at 200 rpm [42]. 

### 2.2. In Silico Analysis

To investigate if orthologs of Cal-Hyr1p exist in *C. auris*, we conducted a sequence homology search and used complimentary homology and energy-based modeling algorithms (Phyre2) [47] to prioritize remote template detection, alignment, 3D modeling, and ab initio protocols. Model refinement was performed with the iTasser server [48]. As a confirmatory measure, additional stochastic modeling was undertaken using the Quark server [49]. Select regions of resulting comparative homologs were then subjected to 3D alignment to identify areas of greatest homology using the Smith–Waterman [50] algorithm, as implemented within Chimera [51]. The top protein models with the highest confidence score for each protein were used for structural similarity determination using the Tm-align tool [52]. An alignment score (Tm-score) of >0.5–1.0 is considered to represent a high degree of similarity in folded protein structure. 

### 2.3. Monoclonal Antibody Generation

We sub-contracted the generation of monoclonal antibodies targeting Hyr1p#5 to ProMab Biotechnologies, Inc. ProMab synthesized the peptide, conjugated with KLH for mice immunization. After three immunizations, mice serum samples were collected and screened for anti-Hyr1p#5 polyclonal antibodies using ELISA plates coated with peptide#5. The mouse with the highest anti-Hyr1p#5 specific antibodies was selected for hybridoma generation. The splenocytes were hybridized with Sp2/0 cells and selected for clones on a selection media. After sub-cloning, 11 clones were selected from a pool of 3000 clones, and their supernatants were collected for analysis using ELISA plates coated with peptide#5 and flow cytometry using *C. albicans* germ tubes. The top 2 clones showing high binding to peptide#5 epitope and *C. albicans* germ tubes were selected for further development. Selected hybridoma clones were cultured in a scaled-up bioreactor and MAbs were purified using protein A/G column, as per the manufacturer’s instructions. The purity of MAbs was confirmed using SDS-PAGE and quantified using UV measurements. After purification, we verified the binding of MAbs to peptide#5 using ELISA, and to *C. albicans* germ tubes and *C. auris* cells using flow cytometry. 

### 2.4. Binding Affinity Determination by Microscale Thermophoresis (MST)

The binding affinity of the MAbs was determined by MST assay using Monolith NT.115pico (NanoTemper Technologies, Munich, Germany). For the target, we conjugated Hyr1p peptide#5 with bovine serum albumin protein and labeled it with NHS-Red (NanoTemper Technologies, Munich, Germany). Briefly, 0.54 µM of the conjugated peptide was mixed with 2.5 µM of NHS-Red labeling dye in a 1:1 ratio, and unbound dye was removed by spinning the mixture in a column provided with the labeling kit. A binding assay was performed in 0.2 mL microtubes by adding 5 nM of the labeled target with a series of 2-fold monoclonal antibody dilutions ranging from 3500 nM to 0.1 nM in Micro-Scale Thermophoresis (MST) assay buffer (50 mM Tris-HCl, pH 7.6, 150 mM NaCl, 10 mM MgCl2, 0.05% Tween-20). This mixture was loaded on low-binding premium capillaries (NanoTemper Technologies, Munich, Germany) and MST signals were measured with 5% LED power and 25 °C fixed temperature. The recorded MST signals of each interaction were plotted against the concentration on one graph. The data were fitted with the help of the quadratic fitting formula (K_d_ formula) derived from the law of mass action. Average binding affinities calculated from three independent experiments were reported for each MAb clone. 

### 2.5. Flow Cytometry

*C. auris* yeast or *C. albicans* germ tube cells were blocked with 1% bovine albumin serum solution (in 1× PBS) at 4 °C for 1 h. After blocking, these cells (2 × 10^6^ cells/tube) were added to 1.5 mL microcentrifuge tubes and centrifuged at 13,000 rpm for 5 min. The pellet was resuspended in 100 μL of HX01 or HX02 or isotype-matched control antibody solution in 1× PBS and incubated for 1 h at room temperature. After the incubation, the cells were washed three times with 1× PBS before adding 100 μL of Alexa Fluor 488-labeled anti-mouse IgG1 detection antibodies (1:100 dilution in PBS, Invitrogen, Waltham, MA, USA). After 1 h of incubation at room temperature, the cells were washed three times and resuspended in 300 μL of PBS. The stained cell suspension was transferred to flow tubes and 20,000 events were acquired using an LSR II flow cytometer (BD Biosciences, San Jose, CA, USA). Flow cytometry data were analyzed using FlowJo software (Version 10).

### 2.6. Biofilm Formation Assay

Biofilms were developed in 96-well polystyrene microtiter plates, as previously described, with some modifications [42]. Briefly, 50 μL of *C. auris* cells (1 × 10^5^ cells/mL in YNB medium) was added to the wells (*n* = 6/test) containing 50 μL (10 µg) HX01 or isotype IgG1 control and incubated at 37 °C for 24 h. The next day, the wells were washed twice with PBS and the extent of biofilm formation was quantified by XTT assay (450 nm). Data are presented as % biofilm reduction ([1-OD450 of wells with HX01/OD450 of wells with isotype IgG2 antibody] × 100) [42,53].

### 2.7. Opsonophagocytic Killing (OPK) Assay

The opsonophagocytic killing assay was based on a modification of a previously used method [42]. Initially, 50 µL (1 × 10^5^ yeast cells) of *C. auris* (CAU-03 or CAU-09) was added into 96-well microtiter plates. To these wells, 50 µL (10 µg) of HX01 MAb or isotype-matched control antibody was added and the plate was incubated at 4 °C for 1 h. Murine macrophages were harvested from the intraperitoneal cavity of 6-to-8-week-old CD-1 mice and washed twice with 1× PBS. This macrophage-enriched cell suspension was pelleted and subsequently resuspended in 1× RPMI (supplemented with 10% fetal bovine serum) media. Macrophage cell suspension density was adjusted appropriately to achieve 2.5 × 10^5^ cells/100 µL/well. The macrophage cells were added to the plate containing *C. auris* (pre-incubated with no antibody, MAb/control antibody solution) at a 1:2.5 ratio of yeast to phagocytes. *C. auris* cells without macrophages and antibody and *C. auris* with macrophages but no antibody served as non-OPK controls. After 2 h of incubation with gentle shaking, replicates (*n* = 5/test) from the wells were quantitatively plated in YPD agar plates. The percent killing of *C. auris* was calculated using the following formula: {(X − Y)/X} × 100, where X = *C. auris* CFU in the absence of antibody and macrophages and Y = *C. auris* CFU with macrophages added in the presence or absence of antibody.

### 2.8. Mice Infection and Treatment

For antibody in vivo efficacy evaluation, we used ICR CD-1 mice immunosuppressed with 200 mg/kg cyclophosphamide intraperitoneal (i.p.) and 250 mg/kg cortisone acetate (sub-cutaneous) injections on day −2, relative to infection. To prevent bacterial superinfection in the immunosuppressed mice, we added enrofloxacin (at 50 μg/mL) to the drinking water on the day of immunosuppression and continued for a week. These mice were infected with *C. auris* (CAU-09) through tail vein injection using 5 × 10^7^ yeast cells/mouse inoculum. For *C. albicans* infection, immunocompetent ICR CD-1 mice were administered 2 × 10^5^ yeast cells/mouse through the tail vein. The infected mice were treated with a 30 µg/mouse dose of either HX01 or IgG1 isotype control antibody on day +1, relative to infection. A repeat dose of the antibodies was administered to mice on day +8, since antibody half-life in mice is expected to be ~4 to 8 days [54,55]. Infected mice were monitored for their survival for 21 days.

For fungal burden determination, mice were infected and treated as above and then euthanized on day 4 post-infection. The mice were weighed before euthanization, followed by the harvesting of kidneys and hearts (primary target organs) [42] for fungal enumeration. Briefly, organs from each mouse were weighed, homogenized, and quantitatively cultured by 10-fold serial dilutions on YPD plates. Plates were incubated at 37 °C for 48 h before enumerating colony-forming units (CFUs)/gram of tissue. Finally, histopathological examination of the kidneys or brains of mice sacrificed on day 4 post-infection were fixed in 10% zinc-buffered formalin, embedded in paraffin, sectioned, and stained with Pacific Acid Schiff (PAS) stain. Stained tissue sections were imaged on Olympus microscopy.

The entire study design is summarized in Figure 1.

### 2.9. Statistical Analysis

Differences in survival studies were analyzed by the non-parametric Log-rank test for overall survival and with Mantel–Cox comparisons for median survival times. All other comparisons were conducted with the non-parametric Mann–Whitney test. *p* values < 0.05 were considered statistically significant. 

## 3. Results

### 3.1. C. auris Has Orthologs of Cal-Hyr1p

*C. auris* has been reported to express orthologs of several *C. albicans* cell wall proteins, including the Als3p and Hyr1p, which are only expressed on *C. albicans* hyphae [28,35]. To investigate if orthologs of Cal-Hyr1p exist in *C. auris*, we conducted sequence homology searches and protein structure modeling and alignment. Consistent with a previous report showing the presence of at least eight Hyr/Iff-like proteins (HlL-1 to 8) in *C. auris* [46], we found at least three cell surface Hyr/Iff-like proteins (HIL-5, 7, and 8) that are universally present in all four clades of *C. auris* and have considerable sequence and structural homology to Cal-Hyr1p (Figure 2). The template modeling and alignment score (Tm-score) of the N-terminal region of Cal-Hyr1p (1–320 amino acids) and its HIL orthologs (HIL-5: 1–366, HIL-7: 1–320, and HIL-8: 1–350 amino acids) in *C. auris* ranged from 0.74–0.85, indicating highly similar protein folding structure. Earlier, we identified a highly immunogenic and surface-exposed epitope on Cal-Hyr1p (Cal-Hyr1p#5 LKNAVTYDGPVPNN) [45]. This epitope was found to be conserved and surface exposed among Cal-Hyr1p and *C. auris* HIL orthologs (Figure 2).

### 3.2. Anti-Cal-Hyr1p#5 Monoclonal Antibody Development 

We contracted anti-Cal-Hyr1p#5 MAb generation to ProMab Biotechnologies, Inc. ProMab provided culture supernatant containing IgG1 isotype MAbs from 11 hybridoma clones (from a pool of 3000 clones). These clones were tested for binding to the original Cal-Hyr1p#5 peptide epitope and the Cal-Hyr1p native protein on C. albicans germ tubes by flow cytometry. All 11 clones showed binding to both Cal-Hyr1p#5 and C. albicans germ tubes (Appendix A). We selected the top 2 clones (8G9D5 (HX01 hereafter) and 6F7C6 (HX02 hereafter) of the IgG1 isotype) that also had cross-reactivity to Gram-negative bacteria, since Cal-Hyr1p#5 is conserved among these bacteria [44,45]. The selected HX01 and HX02 MAb clones were produced and purified from scaled-up hybridoma cultures for further testing (Figure 3A).

First, we determined the binding affinity of HX01 and HX02 MAb clones by microscale thermophoresis (MST). The binding affinity (K_d_) of HX01 and HX02 to Cal-Hyr1p peptide#5 epitope was 21.9 ± 6.0 nM and 40.5 ± 4.9 (mean ± standard deviation), respectively (Figure 3B). Further, we investigated if Anti-Cal-Hyr1p antibodies would recognize *C. auris* using flow cytometry. Binding of the two MAbs to C. albicans was used as a positive control in this binding assay. Clones HX01 and HX02 bound to 28.6% and 24.8% of *C. auris*, and 97.2% and 98.7% of C. albicans pregerminated cells, respectively, whereas an isotype-matched IgG1 did not bind to any of the tested yeasts (Figure 3C). We chose HX01, which has better binding affinity for further in vitro and in vivo testing. Specifically, we tested the binding of HX01 against different clinical isolates of *C. auris* (i.e., CAU-01, CAU-03, CAU-05, and CAU-7) representing different clades and found a strong binding activity against all *C. auris* clades (Appendix A). 

### 3.3. HX01 MAb Inhibits C. auris Biofilm Formation In Vitro and Augments Macrophage Opsonophagocytic Killing (OPK) of C. auris 

*C. auris* HIL proteins orthologs are predicted adhesins and potentially involved in biofilm formation [46]. We assessed if HX01 MAb can block *C. auris* biofilm formation. *C. auris* was incubated in 96-well plates with either HX01 MAb or an isotype-matched IgG1 for 24 h. Next, the plate was gently washed with PBS and biofilm formation was quantified by XTT assay [42]. Compared to wells that contained the pathogen and had no added antibodies, HX01 MAb considerably inhibited *C. auris* biofilm formation (Figure 4A) by 30–40% vs. 10% biofilm inhibition with the isotype-matched IgG1. 

IgG1 antibodies can opsonize and enhance the killing activity of phagocytes. We evaluated the effect of HX01 MAb on the OPK activity of macrophages against *C. auris*. *C. auris* strains CAU-03 and CAU-09 were incubated with HX01 MAb or isotype control IgG1 or no antibody for 1 h, followed by the addition of murine macrophages. *C. auris* alone (*n* = 5 replicates) was also incubated to determine the original CFU. The percentage of killed *C. auris* cells was determined by quantitative culture and presented as % OPK activity. HX01 MAb enhanced murine macrophage OPK activity against both CAU-03 and CAU-09 strains of MDR *C. auris* by 30–35% compared to isotype-matched IgG1 (~75% killing with HX01 vs. ~40% in no antibody or isotype-matched control antibody) (Figure 4B).

### 3.4. HX01 MAb Protects Mice from Hematogenously Disseminated Candidiasis Due to C. auris Infection 

Considering that HX01 MAb binds to *C. auris* and has anti-*C. auris* activity in vitro, we tested the efficacy of HX01 MAb in the mouse infection model. Immunosuppressed outbred CD-1 mice (*n* = 20/group from two independent experiments with similar results) were infected intravenously with a clinical isolate of *C. auris* (CAU-09) resistant to azoles and amphotericin B, and were treated with a 30 µg/mouse dose of HX01 or an isotype-matched IgG1 on days 1 and 8 post-infection. The infected mice were monitored for survival efficacy for 21 days; 55% of HX01-treated mice survived with a median survival time >21 days compared to the isotype-matched IgG1-treated mice that had only 10% survival with 14.5 days median survival time (Figure 5A). 

In a different set of experiments, we infected and treated immunosuppressed mice (*n* = 20/group from two independent experiments with similar results) as above and sacrificed the mice by day +4 post-infection to evaluate the tissue fungal burden in the target organs (kidney and heart) [42]. Mice treated with HX01 had 10 times less (~1.0-log) fungal burden in both the kidney and heart vs. mice treated with isotype-matched IgG1 (Figure 5B). Moreover, histopathological examination of kidneys and hearts collected from mice sacrificed at the same time point of the tissue fungal burden experiment (i.e., day +4 post-infection) corroborated the tissue fungal burden data. Specifically, HX01 treatment resulted in a significant reduction in the number and size of *C. auris* infection foci present in kidneys and hearts compared to foci found in organs harvested from isotype-matched IgG1-treated mice (Figure 6). Collectively, these data clearly show that HX01 significantly protected mice from systemic MDR *C. auris* infection. 

### 3.5. HX01 Does Not Protect Mice from Hematogenously Disseminated Candidiasis Due to C. albicans Infection 

We also tested HX01 against *C. albicans* in an immunocompetent hematogenously disseminated candidiasis mouse model. While 30% of mice treated with HX01 survived the infection by day 21 vs. 0% for isotype-matched control-treated mice, this difference was not significant (*p* = 0.159). Additionally, the median survival time for mice treated with HX01 and isotype-matched control-treated mice was 11 and 9.5 days, respectively (Appendix A).

## 4. Discussion

The emergence of multi-drug resistance in fungal pathogens such as *C. auris* highlights the importance of developing alternative measures to antifungal drug therapy. Previously, we have shown that active and passive vaccination strategies using rHy1p-N as a target antigen results in significant protection of mice from *C. albicans* infections [56,57]. *C. auris* has been reported to express orthologs of several *C. albicans* cell wall proteins, including those exclusively expressed on *C. albicans* hyphae [42]. To investigate if orthologs of Cal-Hyr1p exist in *C. auris*, we conducted in silico protein modeling and homology analysis. We found that *C. auris* contains cell wall proteins that share both sequence and structural similarity with *C. albicans* Hyr/Iff-like (HIL) proteins. The presence of these Cau-HIL proteins has been previously reported [46]. At least three of these HIL proteins have a high Tm score compared with Cal-Hyr1p, are predicted to be surface exposed, and are present among all four clades of *C. auris* [46]. 

Previously, we showed that Cal-Hyr1p has a high structural similarity with Gram-negative bacterial cell wall proteins including outer membrane protein A (OmpA) and filamentous hemagglutinin protein (FhaB) [44,45]. Moreover, we have identified a shared epitope (epitope#5) on Cal-Hyr1p that was also mapped on FhaB and OmpA. Our extensive bioinformatics analyses showed that this epitope#5 is highly immunogenic, surface exposed, and conserved among the three HIL proteins identified to be present in all four clades of *C. auris* (Figure 2). We previously showed that MAbs targeting epitope#5 recognized Gram-negative bacteria and protected mice from pneumonia due to *Acinetobacter baumannii* or *Klebsiella pneumoniae* [44,45]. Here, we show that Cal-Hyr1 epitope#5 is also shared with *C. auris* HIL proteins, and MAbs (HX01 and HX02) targeting this epitope also recognize the cell surface of all four clades of *C. auris.* As expected, the binding of HX01 and HX02 to *C. auris* was less than the binding of these MABs to *C. albicans* pregerminated cells, since these MAbs were raised against *C. albicans* epitope#5. 

The anti-epitope#5 HX01 IgG1 demonstrated significant protective activity against murine *C. auris* hematogenously disseminated infection. This activity was shown in prolonged survival, reduction in tissue fungal burden, and improved organ architecture of mice treated with physiological concentration of HX01 when compared to mice treated with control isotype-matched IgG1. Although detailed studies need to be performed to decipher the mechanism of protection of HX01, it is reasonable to predict that this protection can be due to the ability of HX01 to reduce the ability of *C. auris* to form biofilm. Multicellular communities such as biofilms present in abscesses are thought to protect the pathogen from immune cell recognition/elimination and from antimicrobial drug treatment [58]. Similar to *C. albicans* Als and Iff/Hyr proteins, *C. auris* cell wall orthologs of Als and HIL are predicted to encode proteins that have domains rich in serine/threonine repeats and aggregates forming sequences. Such sequences are important for *Candida* adhesion and biofilm formation. Indeed, HX01 was able to reduce the ability of *C. auris* biofilm formation in vitro and reduce the number and size of abscesses in vivo. However, it is noted from comparative genomics, and our previous work with Als orthologs, that the HIL family members are not the only proteins that can be implicated in adhesion and biofilm formation of *C. auris* [42,46,59,60]. Further, mechanistic studies are needed to decipher the mechanism by which HX01 protects against biofilm formation.

Another potential mechanism of protection of HX01 is the ability of the MAb to enhance phagocyte killing of the yeast. HX01 enhanced the ability of murine macrophages to kill *C. auris* ex vivo by ~2-fold compared to an isotype-matched IgG1 (Figure 4B). This was also evident by the ~1-log reduction in tissue fungal burden of the target organs harvested from mice treated with HX01 vs. those treated with the control isotype-matched IgG1. Importantly, the protective activity of HX01 was seen in a neutropenic mouse model. Although cyclophosphamide treatment results in pancytopenia, it has minimal effect on tissue macrophages, which are derived from embryonic, and not hematopoietic, origin [61]. These results are in agreement with a previous report showing that several fully human MAb targeting Hyr1p bound to *C. auris* and enhanced phagocytosis and clearance of the yeast by mouse macrophages [62].

In addition to targeting *C. albicans* Hyr1p, several MAbs have been tested in preclinical models of *C. auris* infections. These MAbs also target *C. auris* cell wall proteins and have protective efficacy in disseminated candidiasis mouse models. A murine monoclonal antibody 2G8 raised against β-1,3 glucans was shown to bind to multiple pathogenic fungi including *Candida* spp. and *Aspergillus*. A humanized version of this antibody (H5K1) bound to *C. auris,* inhibited the growth of the yeast when used alone, and had a synergistic effect in the time kill curve of *C. auris* when combined with either caspfungin or amphotericin B [63]. In addition, and similar to HX01, H5K1 enabled enhanced phagocytosis of *C. auris* by macrophages [63]. However, no data exist on the efficacy of H5K1 in an in vivo model of infection. 

Another MAb C3.1 that targets β-1,2-mannotriose (β-Man_3_) significantly extended survival and reduced fungal burdens in target organs of *C. auris*-infected mice [64]. Furthermore, two MAbs, 6H1 and 9F2, that target the hyphal wall protein 1 (Hwp1) and phosphoglycerate kinase 1 (Pgk1), respectively, significantly enhanced the survival of mice infected with *C. auris* and reduced fungal burden of targeted organs. Interestingly, a 6H1+9F2 cocktail induced significantly enhanced protection, compared to treatment with MAb individually [64]. It is possible that the addition of HX01 to this cocktail might enhance the protection outcome. 

While HX01 was protective against *C. auris* murine infection, the MAb did not protect against murine *C. albicans* infection. This difference in activity of HX01 is likely due to the difference in the mechanism of protection against these two distinct infections. Although antibodies against certain antigens were shown to be protective in mice infected with *C. albicans* (see below)*,* a protective immune response against *C. albicans* hematogenous infections are known to be reliant on Th1/Th17 immune responses [65]. In contrast, antibodies appear to play a protective role against *C. auris* [42]. 

Protective MAbs against hematogenously disseminated *C. albicans* infection have been reported. For example, a mouse MAb 2G8 (IgG2b) targeting β-glucans protects mice from *C. albicans* hematogenous infection [66]. However, the in vivo efficacy against systemic *C. albicans* infection was not reported. Another MAb B6.1 (murine IgM) targeting *C. albicans* β-1,2-mannotriose administered 1 h post-infection reduced fungal burden by ~28% and enhanced survival time. However, administration of B6.1 to mice 2 h post-infection was not protective. Interestingly, administration of B6.1 with non-therapeutic doses of amphotericin B enhanced survival time of mice beyond what is seen by treatment of B6.1 alone [67]. Furthermore, treating mice with anti-aspartic proteinase antibodies protected mice from *C. albicans* infection [68]. Mycograb, a recombinant scFv anti-HSP90 (chaperone protein of *C. albicans*), has shown protective efficacy against *C. albicans* [69]. Mycograb was tested against systemic candidiasis in clinical trials, but was not approved by regulatory agencies due to product safety and quality issues.

Therapeutic MAbs targeting different cell wall proteins and polysaccharides have also been reported against other fungi in murine models. Murine MAbs targeting capsular polysaccharide purified from *Cryptococcus neoformans* were found to increase the survival of mice and reduce fungal burden in target organs [70]. Another murine IgG1 MAb, 18B7, which targets *C. neoformans* polysaccharide, has shown treatment benefits in clinical trials [71]. Murine IgG1 MAb (P6E7) targeting 70 kDa glycoprotein of *Sporothrix* spp. reduced fungal burden in the spleen and liver of mice infected with *S. schenchii* [72]. Similarly, a cocktail of murine IgG1 MAbs (B7D6 and C5F11) targeting 70 kDa glycoprotein of *Paracoccidioides brasiliensis* was protective in a murine model of paracoccidioidomycosis [73]. Murine MAb against histone H2B of *Histoplasma capsulatum* prolonged survival, reduced fungal burden, and decreased inflammation in a mouse model of infection [74]. MAb R-5 targeting enolase (a 48 kDa protein) of *Aspergillus* spp. reduced fungal burden by ~85% and significantly improved mice survival [75]. Finally, C2 mouse MAb (IgG1) targeting the invasin, CotH3 protein, was protective against murine mucormycosis due to several Mucorales fungi [76]. Several of these MAbs were also tested in combination with antifungal drugs in a murine model of systemic mycoses and found to have significantly enhanced protective efficacies compared to monotherapy. The protective mechanism of these MAbs include enhancement of opsonophagocytic killing, prevention of host tissue invasion and biofilm formation, direct fungal killing, and growth inhibition, as well as enhancement of immune functions. Collectively, these studies provide compelling evidence to make a case for therapeutic MAb against systemic fungal infections, including those caused by *C. auris*.

## 5. Conclusions

In conclusion, we showed that MDR *C. auris* contains HIL proteins, and those proteins can be an ideal target for immunotherapeutic antibodies. Considering the scarcity of effective antifungal drugs and the lack of other alternative treatment options, a HIL protein-based MAb approach is likely to prove highly effective in managing *C. auris* infections. This application of antibodies can be either for prophylactic measure in high-risk patients, since MAbs usually have a long half-life in humans [77], or therapeutically with antifungal therapy. Our future studies will focus on the isolation of a humanized version of HX01, the determination of its toxicity, and the development of a viable cell line that will produce the humanized MAb in commercial quantities to enable clinical trial testing. 

## Figures and Tables

**Figure 1 jof-09-00103-f001:**
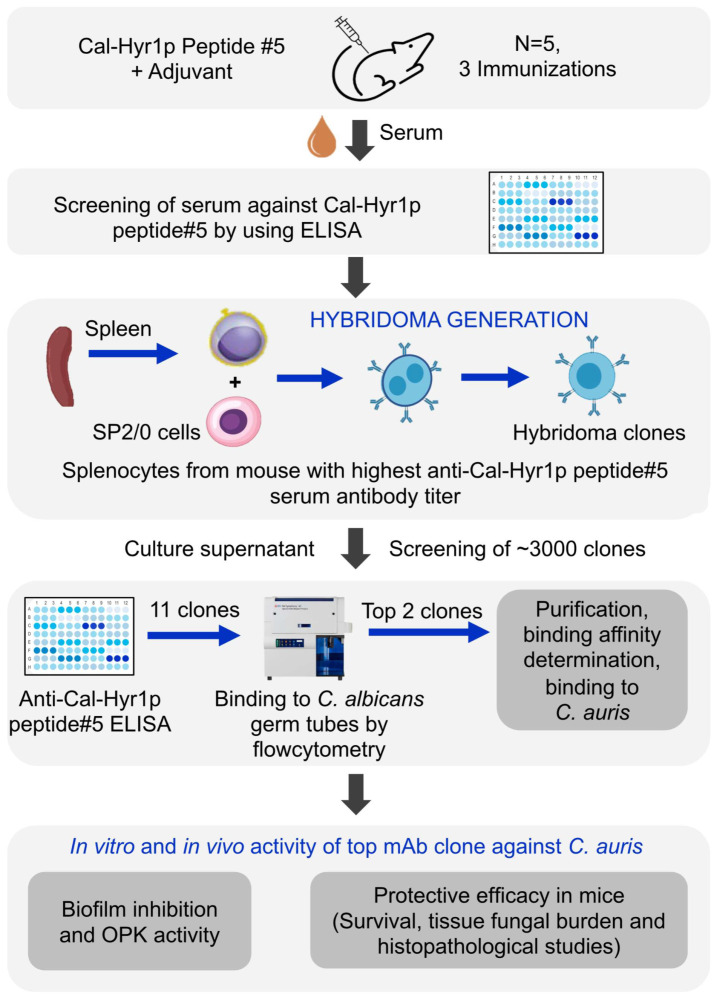
Overview of the Anti-Cal-Hyr1p peptide#5 MAb development and their in vitro and in vivo testing.

**Figure 2 jof-09-00103-f002:**
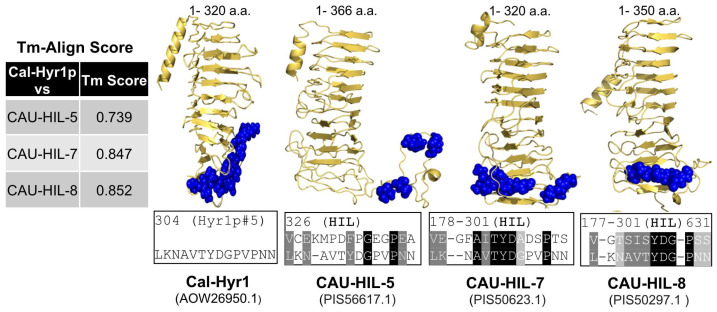
Cal-Hyr1 shows high structural similarity with Hyr1/Iff-like proteins (HIL) of *C. auris*. Cal-Hyr1 was used to blast search similar proteins in *C. auris*. Top hits that also contain high sequence identity with Cal-Hyr1 antigenic peptide#5 epitope were selected to generate a 3D model using the iTasser server. Protein models containing cell surface-exposed N-terminal regions of each protein were aligned against the Cal-Hyr1p model and Tm alignment scores were generated (Table). A Tm score of >0.5 indicates proteins with similar folding. Cal-Hyr1p peptide#5 homologous regions are visualized with blue spheres. Sequence homology between Cal-Hyr1 peptide#5 and *C. auris* HIL proteins is shown under each model.

**Figure 3 jof-09-00103-f003:**
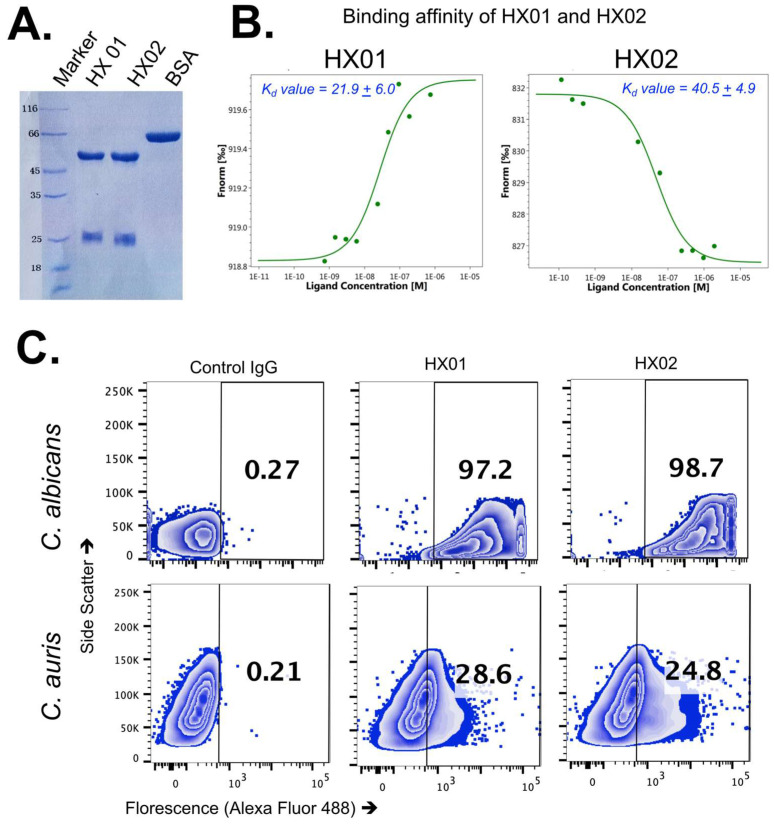
Binding activity of purified anti-Cal-Hyr1 MAbs. (**A**) An amount of 3 µg of purified HX01, HX02, or BSA was loaded on the SDS page to determine its purity. (**B**) HX01 and HX02 binding affinities were determined against Hyr1 peptide#5 conjugated with BSA. Antibody concentrations are plotted against fluorescent signal captured. Data are the average of three independent experiments. (**C**) Each organism (pre-germinated C. albicans SC5314, *C. auris* (CAU-09)) had 2 × 10^6^ cells incubated with 10 μg/mL of HX01 and HX02 MAb or an isotype-matched control IgG1. Bound antibodies were detected by anti-mouse IgG1 labeled with Alexa Fluor 488. The extent of binding was quantified by flow cytometry. Data are presented in a zebra plot highlighting the percentage of each organism that was bound by each Ab.

**Figure 4 jof-09-00103-f004:**
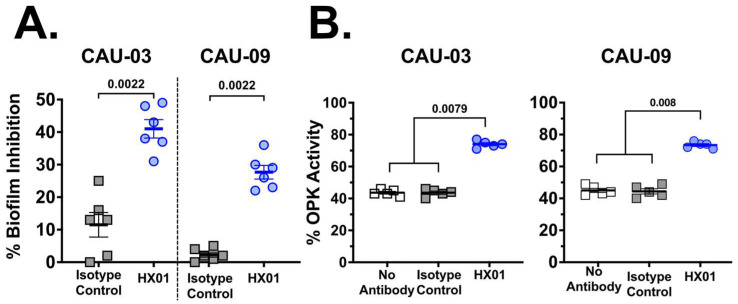
HX01 MAb inhibits *C. auris* biofilm formation and enhances OPK activity of murine macrophages. (**A**) *C. auris* (0.1 million cells/well of CAU-03 and CAU-09 isolates) biofilm formation was evaluated in 96-well plates in the presence of HX02 or an isotype-matched control IgG. HX01 significantly reduced *C. auris* biofilm formation compared to control IgG (*n* = 6). Percent biofilm inhibition was determined by comparing wells with *C. auris* with no antibody added. (**B**) OPK of *C. auris* (0.1 million cells/well) by murine macrophages (0.25 million cells/well) was evaluated in the presence of HX01 or an isotype-matched control IgG. HX01 significantly enhanced yeast OPK compared to control antibody (*n* = 5). Percent killing of *C. auris* was calculated by formula: {(X − Y)/X} *×* 100, where X = *C. auris* CFU in the absence of antibody and macrophages and Y = *C. auris* CFU with macrophages added in the presence or absence of antibody. Data are representative of two independent experiments. Statistical significance was determined by the Mann–Whitney Test. *p* values < 0.05 were considered statistically significant.

**Figure 5 jof-09-00103-f005:**
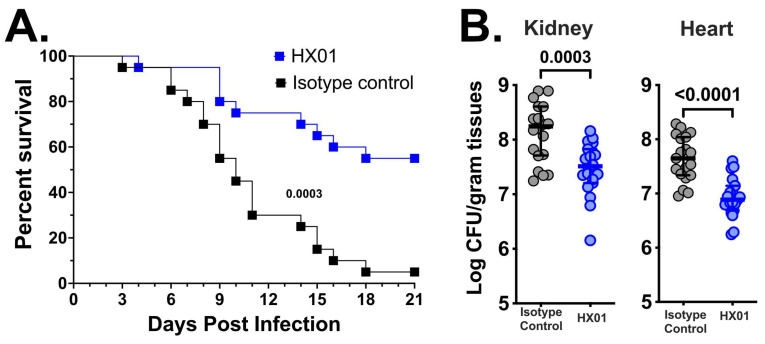
HX01 MAb protects mice from *C. auris* infection. CD-1 mice (*n* = 20/group) were immunosuppressed with cyclophosphamide and cortisone acetate. Two days after the immunosuppression, mice were infected intravenously with *C. auris* CAU-09 (5 × 10^7^ cells). After 24 h of infection, mice were randomized and divided into two treatment groups of 30 µg/mouse of either HX01 MAb or an isotype-matched control IgG1 (i.p.). Mice survival (compared by Log-rank Test) (**A**) or tissue fungal burden (determined by quantitative culturing on day +4, relative to infection, and analyzed by the Mann–Whitney Test) of the kidney and heart (**B**) were evaluated in separate experiments. Data in (**A**,**B**) were combined from two independent experiments. *p* values < 0.05 were considered statistically significant.

**Figure 6 jof-09-00103-f006:**
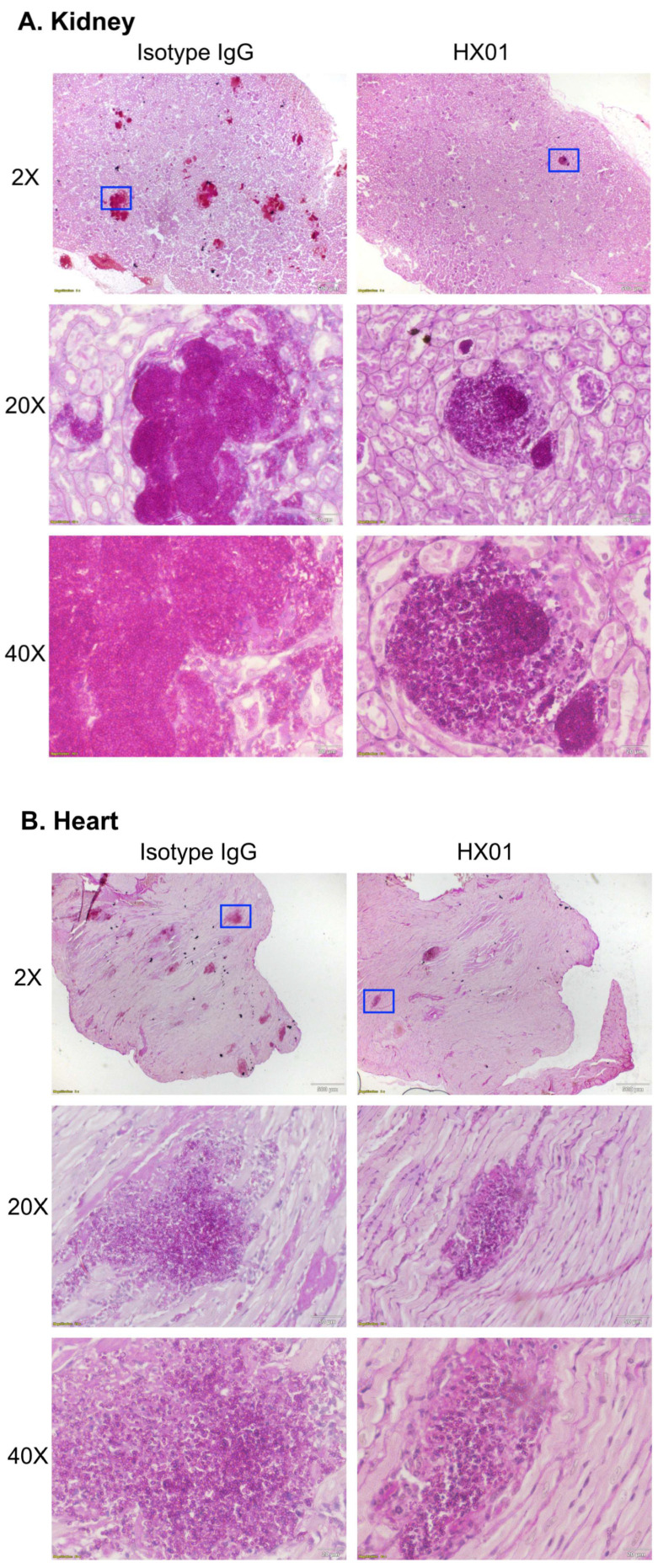
Histopathology of *C. auris*-infected mice treated with HX01 MAb. Immunosuppressed CD-1 mice were infected intravenously with *C. auris* CAU-09 (5 × 10^7^ cells) and treated with 30 µg/mouse of HX01 MAb or an isotype-matched control IgG1 (i.p.) on day +1 post-infection. Mouse kidney (**A**) and heart (**B**) histopathology sections were stained with PAS and imaged using Olympus bright field microscopy. Blue boxes in top panels represent areas magnified at 20× and 40× in the middle and lower panels, respectively.

## Data Availability

All data supporting the findings in this manuscript are available upon request.

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
