# Peer review of "Protective Efficacy of Anti-Hyr1p Monoclonal Antibody against Systemic Candidiasis Due to Multi-Drug-Resistant Candida auris"

_jof, 2023, doi:10.3390/jof9010103_

Round 1

Reviewer 1 Report

Shakti Singh et al., described a new monoclonal antibody against C. albicans Hyphal-regulated protein (Cal-18 Hyr1p) that recognized ortholog protein in C. auris (Cau-HILp). This mAb prevented biofilm formation, enhanced killing by macrophages and protected 55% of mice against lethal systemic C. auris. The results are interesting and provide new information about monoclonal antibody therapy against systemic fungal infections.

1-    Whereas other species of the candida genus are important (C. albicans, C. parapsilosis, etc.). Why did the authors not at least include the data with C. albicans. mAbs HX01 and HX02 appear to have activity against several species of the Candida genus.

2-    mAbs HX01 and HX02 bound to 28.6% and 24.8% of C. auris, and 97.2% and 98.7% of C. albicans. Does this reduction occur by the amount of expressed protein or by antibody affinity?

3-    Authors showed that HX02 MAb inhibits C. auris biofilm formation and enhanced phagocytosis and macrophage killing. Why did the authors not analyze the immune activity of peritoneal macrophages?

4-    The animals were treated with a dose of 30 μg/mouse of HX02 on days 1 and 8 post-infection. Did the authors perform a kinetics of circulating antibody activity?

5-    Did the authors associate the treatment of C. auris with mAbs and echinocandins? If not, what do the authors think about this association?

6-    Despite the results presented in the article, in my view the discussion could have comparisons with other protective monoclonal antibodies against systemic fungal infections.

Author Response

Response to reviewer # 1

General comment: The results are interesting and provide new information about monoclonal antibody therapy against systemic fungal infections.

We thank the reviewer for the constructive comment.  While revising the manuscript we noticed that we inadvertently switched the names of HX01 with HX02 of the MAb.  The clone that had the higher affinity to peptide#5 and was further tested in vitro and in vivo is clone HX01 and not clone HX02.  While this has no effect on the scientific content of the manuscript, we corrected these inadvertent errors throughout the revised manuscript in Track changes.

Specific Comments:

1-    Whereas other species of the candida genus are important (C. albicans, C. parapsilosis, etc.). Why did the authors not at least include the data with C. albicans. mAbs HX01 and HX02 appear to have activity against several species of the Candida genus.

We have tested the activity of HX01 against murine C. albicans hematogenous infection. While 30% of mice treated with HX01 survived the infection by day 21 vs. 0% for isotype-matched control-treated mice, this difference was not significant (P=0.159). Additionally, the median survival time for mice treated with HX01 and isotype-matched control-treated mice was 11 and 9.5 days, respectively. This data is now introduced in the manuscript as Figure S3 and now is discussed in the manuscript (Lines 370-377). We believe that the mechanism of protection against murine C. albicans and C. auris hematogenous infections are different with the former being heavily reliant on Th1/Th17 immune response (this is supported by multiple publicans from our laboratory and others) and the latter for which antibodies play a pivotal role in protection (Singh et al. PLoS Pathogens 2019 [reference 42 in the manuscript]).    

2-    mAbs HX01 and HX02 bound to 28.6% and 24.8% of C. auris, and 97.2% and 98.7% of C. albicans. Does this reduction occur by the amount of expressed protein or by antibody affinity?

The Hyr1p and its HIL orthologs are not identical and belong to large families of C. albicans and C. auris proteins, respectively. It is hard to compare the expression of these two proteins since HX01 and HX02 are likely to bind to other family members. The data presented rather show the percent of yeast cells that have been stained by each of the MAb clones.  This is now clearly stated in the manuscript (Lines 286-288) and figure legend.

3-    Authors showed that HX02 MAb inhibits C. auris biofilm formation and enhanced phagocytosis and macrophage killing. Why did the authors not analyze the immune activity of peritoneal macrophages?

As stated above, this data is for HX01. As mentioned in the Materials and Methods section, we used peritoneal macrophages to access the OPK activity of HX01 ex vivo. We now introduce a revised figure for which the activity of peritoneal macrophages without any added antibodies is included and %OPK activity is calculated based on CFU counts with no macrophage or antibody added (formula to calculate the % killing is modified accordingly in the method section Line 193-196. Additional studies to decipher the mechanism of protection in vivo are designed for a follow up manuscript.

4-    The animals were treated with a dose of 30 μg/mouse of HX02 on days 1 and 8 post-infection. Did the authors perform a kinetics of circulating antibody activity?

We did not perform kinetics on this particular MAb since it is known that half-life of antibodies in mice is about 4-7 days (Waldmann and Strober Prog. Allergy 1969;13:1-110 and Seijsing et al. Sci Rep 2018;8:5141). This is also our experience with C2 MAb and its humanized version (LC3HC3) for which are both protective against mucormycosis (please see below our unpublished data).

5-    Did the authors associate the treatment of C. auris with mAbs and echinocandins? If not, what do the authors think about this association?

We thank the reviewer for this excellent comment.  We have done this with the C2 MAb against mucormycosis (Gebremariam et al. Science Advances 2019; 5:eaaw1327) and showed synergy of the MAb with Polyenes and Azoles.  We intend to conduct similar future studies with echinocandins with HX01.

6-    Despite the results presented in the article, in my view the discussion could have comparisons with other protective monoclonal antibodies against systemic fungal infections.

We agree with the reviewer. We have included a section on MAb protection in other fungal infections (Lines 465-502).

We thank the editor and the reviewer for the time and many insightful comments! We hope that our manuscript is now acceptable for publication.

Reviewer 2 Report

Minor Comments

In this research article, the authors investigate a highly immunogenic and surface-exposed epitope that is conserved between Cal-Hyr1p and Cau-HILp. An ani-Hyr1p monoclonal antibody can cross-recognize C. auris in vitro and protect mice from C. auris hematogenously disseminated infection. The manuscript is well-written, and the experimental design and data analysis are robust.

Point 1: Sentence 160, Control and incubated at 37°C for 24 hours at 37°C; correct it by removing the extra at 37°C.

Point 2: Figure 1 is ok, would be better to draw one extra figure, a detailed experimental workflow. Otherwise, it would be difficult for the reader to capture the overall picture of the study.

Point 3: It would be better to show data if there is any significance among HX01 and HX02 compared to the control in the following experiments.

1.     %Biofilm inhibition, 2. % OPK activity 3. Survival, CFUs 4. Histopathology

Point 4: Authors need to explain for how many days enrofloxacin treatment was given to mice in the drinking water. Why specifically the two organs (kidneys and heart) chosen for fungal burden determination after mice infection.

Overall, I could not fault the experiments or interpretation. However, future mechanistic investigations would be informative.

Good Luck

Author Response

General comment: The manuscript is well-written, and the experimental design and data analysis are robust.

We thank the reviewer for the constructive comment. While revising the manuscript we noticed that we inadvertently switched the names of HX01 with HX02 of the MAb.  The clone that had the higher affinity to peptide#5 and was further tested in vitro and in vivo is clone HX01 and not clone HX02.  While this has no effect on the scientific content of the manuscript, we corrected these inadvertent errors throughout the revised manuscript in Track changes.

Specific Comments:

Point 1: Sentence 160, Control and incubated at 37°C for 24 hours at 37°C; correct it by removing the extra at 37°C.

This is now fixed.

Point 2: Figure 1 is ok, would be better to draw one extra figure, a detailed experimental workflow. Otherwise, it would be difficult for the reader to capture the overall picture of the study.

We have now included a Figure as requested by the reviewer (Figure 1).

Point 3: It would be better to show data if there is any significance among HX01 and HX02 compared to the control in the following experiments.

  1. %Biofilminhibition, 2. % OPK activity 3. Survival, CFUs 4. Histopathology

We have not conducted experiments with HX01 since our criteria of selection was based on the binding affinity of the MAb to the peptides that they were raised against. HX01 showed twice as much better binding affinity. Hence, it was selected to more detailed functional and protective activity.

Point 4: Authors need to explain for how many days enrofloxacin treatment was given to mice in the drinking water. Why specifically the two organs (kidneys and heart) chosen for fungal burden determination after mice infection.

Enrofloxacin was started from the day of immunosuppression on Day -2, relative to infections and continued for a week. The two organs were selected for further analysis because they are the primary target organs for this model (Singh et al.  PLoS pathogens 2019 [reference 42 in the manuscript]). This is now stated in the Material and Methods section (Lines 213-214).

Overall, I could not fault the experiments or interpretation. However, future mechanistic investigations would be informative.

We agree with the reviewer and that is our intention for the follow up manuscript.

We thank the editor and the reviewer for the time and many insightful comments! We hope that our manuscript is now acceptable for publication.
